# Alcohol Drinking Patterns and Laboratory Indices of Health: Does Type of Alcohol Preferred Make a Difference?

**DOI:** 10.3390/nu14214529

**Published:** 2022-10-27

**Authors:** Onni Niemelä, Mauri Aalto, Aini Bloigu, Risto Bloigu, Anni S. Halkola, Tiina Laatikainen

**Affiliations:** 1Department of Laboratory Medicine and Medical Research Unit, Seinäjoki Central Hospital and Tampere University, 60220 Seinäjoki, Finland; 2Department of Psychiatry, Seinäjoki Central Hospital, Tampere University, 33100 Tampere, Finland; 3Center for Life Course Health Research, University of Oulu, 90570 Oulu, Finland; 4Infrastructure of Population Studies, Faculty of Medicine, University of Oulu, 90570 Oulu, Finland; 5Department of Public Health Solutions, National Institute for Health and Welfare (THL), 00271 Helsinki, Finland; 6Institute of Public Health and Clinical Nutrition, University of Eastern Finland, 70210 Kuopio, Finland; 7Joint Municipal Authority for North Karelia Social and Health Services, 80210 Joensuu, Finland

**Keywords:** beer, binge drinking, ethanol, harm reduction, liquor, wine

## Abstract

Although excessive alcohol consumption is a highly prevalent public health problem the data on the associations between alcohol consumption and health outcomes in individuals preferring different types of alcoholic beverages has remained unclear. We examined the relationships between the amounts and patterns of drinking with the data on laboratory indices of liver function, lipid status and inflammation in a national population-based health survey (FINRISK). Data on health status, alcohol drinking, types of alcoholic beverages preferred, body weight, smoking, coffee consumption and physical activity were recorded from 22,432 subjects (10,626 men, 11,806 women), age range 25–74 years. The participants were divided to subgroups based on the amounts of regular alcohol intake (abstainers, moderate and heavy drinkers), patterns of drinking (binge or regular) and the type of alcoholic beverage preferred (wine, beer, cider or long drink, hard liquor or mixed). Regular drinking was found to be more typical in wine drinkers whereas the subjects preferring beer or hard liquor were more often binge-type drinkers and cigarette smokers. Alcohol use in all forms was associated with increased frequencies of abnormalities in the markers of liver function, lipid status and inflammation even at rather low levels of consumption. The highest rates of abnormalities occurred, however, in the subgroups of binge-type drinkers preferring beer or hard liquor. These results demonstrate that adverse consequences of alcohol occur even at moderate average drinking levels especially in individuals who engage in binge drinking and in those preferring beer or hard liquor. Further emphasis should be placed on such patterns of drinking in policies aimed at preventing alcohol-induced adverse health outcomes.

## 1. Introduction

Excessive alcohol drinking is strongly associated with morbidity and mortality throughout the world. However, the relationships between the amounts and patterns of drinking and health status still remain unclear [1,2,3,4,5]. There has also been a longstanding debate on possible differences between the health issues brought about by different types of alcoholic beverages.

Recent studies have indicated that a causal relationship exists with cumulative ethanol intake and adverse health effects when weekly regular alcohol drinking exceeds 100 g of alcohol [6,7,8,9]. However, due to the fact that ethanol intake exerts toxic effects through a wide variety of biochemical pathways, which also take place in an age- and sex-dependent manner, there may be significant variation in the dose–response relationships and individual risks of disease. Independent of the average drinking levels, binge drinkers seem to be at a higher risk of alcohol problems than those with regular consumption [9,10,11]. Binge drinking, which is typically characterized by repeated bouts of heavy drinking in amounts exceeding 60 g of alcohol (men) or 40 g (women) on each episode [12,13], leads to high blood alcohol levels in a repetitive manner and may thus be expected to create distinct types of health risks through generation of toxic alcohol metabolites, stimulation of inflammation, oxidative stress and lipid peroxidation [14,15,16,17,18].

While several lines of previous evidence have suggested that individuals consuming low to modest amounts of wine show a decreased risk for cardiovascular diseases, a great deal of controversy exists in defining safe levels of alcohol drinking [19,20,21,22]. Recent meta-analyses have concluded that low-volume alcohol drinking shows no net mortality benefits [22]. As yet, only limited information has, however, been available on the comparisons of the drinking patterns and the characteristics of alcohol-related health effects in individuals preferring various types of specific alcoholic beverages. Therefore, the present work was set out to investigate the relationships between alcohol use and laboratory indices of health in a large population-based sample of individuals classified according to the amounts and patterns of drinking and the dominant alcoholic beverage type in their consumption.

## 2. Materials and Methods

### 2.1. Study Design, Data Sources and Participants

Data were collected from a cross-sectional population health survey (The National FINRISK Study) [23] carried out in Finland every five years since 1972 [23]. In this work, the data from surveys between 1997 and 2007 representing an age- and gender stratified random sample drawn from the population register were used as previously described [6,23,24]. Clinical examinations included anthropometric measures, laboratory tests and detailed questionnaires on the amounts and patterns of alcohol intake, types of alcoholic beverages consumed, current health status, diet, smoking, physical activity, medical history and socioeconomic factors [25]. The total study sample consisted of 22,432 apparently healthy individuals: 10,626 men, 11,806 women (mean age 49 ± 13 years, range 25–74 years). The participation rates varied from 52.6% to 73.4% as percentages of individuals who both completed the questionnaires and attended the medical examination. The study excluded individuals with, any apparent clinical signs of liver disease, ischemic heart or brain disease or active infection at the time of the study. The medications reported by the present study subjects included occasional analgesics (22.8%), antidepressants (4.7%), sedatives (5.9%), inhaled asthma medication (5.7%), antihistamines (3.0%) and low-dose acetylsalicylic acid (10.2%).

Data on alcohol consumption were based on self-reports, which were recorded systematically from the past 12 months and past weeks prior to blood sampling. The structured questionnaires used for this purpose covered the types of beverages consumed, the frequency of consumption, and the total amounts of ethanol-containing drinks. The amount of ethanol in different beverage types was quantitated in grams of ethanol based on defined portion sizes as follows: wine 12 g (12 cL), regular beer 12 g (1/3 L), strong beer 15.5 g (1/3 L), long drink 15.5 g (1/3 L), cider 12 g (1/3 L) and spirit 12 g (4 cL), The participants were further classified according to the type of alcoholic beverage preferred (>50% of total consumption) into the following categories (wine, beer, cider or long drink, spirits or mixed type). The mixed group consisted of individuals in whom none of the above specific types of beverages exceeded 50% of the total consumption. The main characteristics of the subjects in each subgroup are summarized in Table 1. The alcohol drinkers were further classified according to the pattern of drinking to either regular drinkers or binge drinkers. Data on regular alcohol consumption from the past 12 months prior to sampling were used to categorize the material to i: abstainers, ii: moderate drinkers (≤7 drinks per week for women or ≤14 drinks per week for men) or iii: heavy drinkers, who exceeded 7 drinks per week (women) or 14 drinks per week (men) [26]. Binge drinking was defined as a pattern characterized by occasional bouts of heavy drinking at least once a month in amounts exceeding 60 g (men) or 40 g (women) on each occasion typically leading to blood alcohol levels above 0.8 per mill, as previously recommended [12,13].

Body weight and height were determined to the nearest 0.1 kg and 0.1 cm, respectively. Body mass index (BMI, kg/m^2^) was subsequently calculated as a measure of relative body weight. Waist circumference was measured to the nearest 0.5 cm between the lowest rib and the iliac crest while exhaling. Data on smoking and coffee consumption were recorded with a set of standardized questions and expressed as the amounts of cigarettes per day and as the intake of standard servings of coffee (cups) per day, respectively. Leisure-time physical activity and the number of physical exercises with intensity leading to shortness of breath or sweating were registered using structured questionnaires, as previously described [6].

The ethical approval for the study was received from the Coordinating Ethics Committee of the Helsinki and Uusimaa Hospital District (1997:38/96; 2002:87/2001; 2007:229/EO/2006). All surveys were conducted in accordance with the Declaration of Helsinki according to the ethical rules of the National Public Health Institute.

### 2.2. Laboratory Analyses

Serum alanine aminotransferase (ALT) and gamma glutamyl transferase (GGT) activities were measured by standard clinical chemical methods on an Abbott Architect clinical chemical analyzer (Abbott Laboratories, Abbott Park, IL, USA) as described previously [6,15]. The measurements of C-reactive protein (CRP) were carried out using a latex high-sensitivity immunoassay (Sentinel Diagnostics, Milan, Italy) on Abbott Architect c8000 immunochemistry analyzer. The levels of total cholesterol, high-density lipoprotein (HDL), low-density lipoprotein (LDL) and total triglycerides were analyzed by standard enzymatic methods. Blood cell counts were measured from one regional subsample of 1715 participants using routine automated hematological assay systems. The cut-offs for the normal limits in these laboratory parameters were as follows: ALT (50 U/L men; 35 U/L women), GGT (60 U/L men; 40 U/L women), CRP (3.0 mg/L), cholesterol (5 mmol/L), HDL cholesterol (1.0 mmol/L men, 1.2 mmol/L women), LDL cholesterol (3.0 mmol/L), triglycerides (1.7 mmol/L), mean corpuscular volume of erythrocytes (MCV) 96 fl.

### 2.3. Statistical Methods

The study variables are presented using means and standard deviations (SDs) or frequencies and percentages. Continuous variables were compared by independent samples t-test and categorical variables by chi-square test. For comparisons between alcohol consumers in general and subgroups preferring different types of alcoholic beverages we used z-test for two proportions. A multivariate binary logistic regression analysis was used to estimate the relative risks of abnormal biomarker levels associated with the dominant type of alcoholic beverage using age, BMI, smoking, physical activity and coffee consumption as covariates. The distribution of findings exceeding the cut-offs for the biomarkers were examined by chi-square test for trend in ordered study groups from low to average drinking levels and binge-type patterns as follows: (1) moderate average drinking without binge-type pattern, (2) moderate average drinking with binge-type pattern, (3) heavy average drinking without binge-type pattern and (4) heavy average drinking with binge-type pattern. Statistical comparisons were performed exploratively without correction for multiple comparisons to reveal tendencies. Correlations between the study parameters were calculated using Spearman’s rank correlation coefficients. For the analyses, IBM SPSS Statistics 28.0 (Armonk, NY: IBM Corp.) was used. A *p*-value < 0.05 was deemed statistically significant.

## 3. Results

The present study population consisted of 22,432 individuals (10,626 men, 11,806 women) of whom 14,212 (7580 men, 6632 women) (63.4%) were alcohol consumers. The main characteristics of the participants classified to abstainers, alcohol consumers and subgroups according to the levels and patterns of drinking are summarized in Table 1. The data in the groups of alcohol consumers, classified according to the dominant type of alcohol-containing beverages that they prefer, are also shown. In the demographic characteristics, alcohol consumers were younger than abstainers (*p* < 0.001 for both men and women). Within the population of alcohol consumers, those preferring beer or cider/long drinks were found to be younger (*p* < 0.001 for both men and women) and those preferring wine or hard liquor older than the population of alcohol consumers in general (*p* < 0.001 for both men and women).

In the total population of alcohol consumers, moderate average-level drinking was markedly more common in wine drinkers whereas the subjects preferring beer or hard liquor showed higher frequencies of binge-type drinking (Table 1). The latter groups were also found to smoke significantly more often than abstainers, alcohol consumers in general, or wine drinkers (*p* < 0.001 for all comparisons). The subgroup of those preferring hard liquor typically showed higher BMI and waist circumference than abstainers or alcohol consumers in general (*p* < 0.001 for both comparisons). Coffee consumption was highest in those preferring beer or hard liquor (*p* < 0.001) and lowest in wine drinkers (*p* < 0.001) (Table 1). In men, those preferring wine also showed the highest levels of physical activity (*p* < 0.01), whereas the lowest levels were registered from the groups representing drinkers of beer and cider or long drinks. In women, the levels of physical activity were rather similar between the study groups.

Table 2 shows the incidences of abnormal values in various biomarkers of liver function, inflammation and lipid status in the various study groups. The data for MCV, a laboratory indicator of long-term alcohol drinking, which were available from a subsample of 1715 participants (1115 men, 600 women), are also shown. In comparisons with the group of abstainers, the abnormalities in the biomarker status were significantly more common in alcohol consumers. The highest rates of abnormalities occurred in the group preferring hard liquor, who also showed higher incidences of abnormal GGT, total cholesterol, HDL-cholesterol, LDL-cholesterol, triglycerides, CRP and MCV than the total population of alcohol consumers in general (Table 2).

Table 3 demonstrates the multivariable relative risks for abnormal biomarker status in alcohol consumers according to dominant alcoholic beverage type as adjusted for age, BMI, physical activity, smoking and coffee consumption. In comparisons with abstainers, the most striking associations with alcohol use and abnormal liver enzymes, LDL-cholesterol, triglycerides and MCV levels were found to occur in men preferring beer or spirits, whereas women were found to be less sensitive to aberrations in lipid status. In both sexes, alcohol consumption was found to be associated with a decreased risk for abnormal HDL-cholesterol.

Table 4 summarizes the data on the trends of abnormal biomarker findings in the ordered subgroups of men and women with or without binge-type pattern of drinking and with either moderate or heavy average levels of drinking. The most significant trends for increased liver enzymes, abnormal lipid status and elevated CRP, a biomarker of inflammation, were observed in those representing drinkers of beer or hard liquor (Table 4). Gender-dependent variation was also observed in the magnitude of the differences with more striking ALT and CRP findings being observed in men. In additional analyses of correlations between the frequency of binge drinking episodes in alcohol consumers from the past one year, the strongest associations were observed between GGT (*r_s_* = 0.234, *p* < 0.001) in men preferring spirit and between MCV in men (*r_s_* = 0.258, *p* < 0.001) and women (*r_s_* = 0.270, *p* < 0.001) preferring beer.

## 4. Discussion

Although alcohol consumers are known to be at a higher risk for developing health problems, relatively little has been known on the specific characteristics of health effects induced by different patterns of drinking or differences in the metabolic consequences brought about by different types of alcoholic beverages. This population-based study of individuals with detailed structured information on the amounts and patterns of alcohol use and health status indicates that independent of the average levels of drinking, distinct features in the patterns of drinking may lead to significant variation in the status of biomarkers predicting adverse health outcomes. Our data also support the view of notable variations in lifestyle-related individual characteristics between subjects preferring different types of alcoholic beverages.

Based on current findings it is possible that the patterns of drinking and beverage-specific differences may also explain previous observations on the dose–response relationships between alcohol consumption and attributable health outcomes [1,9]. Several lines of evidence have suggested that drinking wine in light to moderate amounts may be associated with good benefits of cardiovascular health [19,21,27,28,29]. However, such findings have been reported primarily from samples representing societies with a low prevalence of binge drinking and populations following Mediterranean diets. A number of other studies from different types of cohorts have indicated that no amount of alcohol is safe [9,20,22,30,31,32,33,34]. In Finland, a relatively large proportion of alcohol consumption has traditionally consisted of distilled beverages or beer with a high prevalence of binge-type drinking [35]. The present data suggest that such drinking patterns may also be linked with a significant overrepresentation of health problems.

While the primary mechanisms underlying the present observations remain unclear at this time it should be noted that consumption of distilled beverages with high alcohol by volume content may typically produce higher peak blood alcohol levels and excessive amounts of toxic alcohol metabolites as compared to the levels typically achieved by other types of alcohol [36,37]. There may also be differences in the caloric contents of the different types of beverages such that drinking beer may lead to markedly higher caloric intake than corresponding doses of wine [38]. Interestingly, in the present material, the responses in ALT, which may be considered a biomarker of liver metabolic functioning, was most frequently elevated in binge-type drinkers of beer (Table 4). The relative differences between wine and other types of alcoholic beverages may also be explained by factors unrelated to ethanol, such as beverage polyphenol content or beverage-specific effects on the gastrointestinal bacterial flora, which in turn may also be associated with the individual status of low-grade inflammation [39,40,41]. Here, the responses in CRP, a biomarker of inflammation, were found to occur in a more sensitive manner in drinkers of beer or spirits, whereas wine drinkers showed lower odds for increased CRP levels supporting the view of a greater anti-inflammatory action for wine [41,42]. It should further be noted that the responses in the subsample of MCV analyses (*n* = 1715) were most pronounced in drinkers of hard liquor. Previously, elevated MCV levels in heavy drinkers have also been linked with a risk for upper gastrointestinal tract carcinogenesis where high local levels of acetaldehyde, the first metabolite of ethanol, as typically achieved by ingestion of distilled alcohol products, seem to play a pivotal role [43,44,45]. Due to the smaller number of observations in this sample, the findings should, however, be interpreted with caution at this time.

The present data further indicate that binge-type drinking patterns appear to be most typical among drinkers of beer and hard liquor. While as yet relatively little has been known on the differences between the specific clinical features induced by binge-type drinking or regular alcohol consumption, repeated episodes of binge drinking may obviously be expected to intensify the damage resulting from high blood alcohol contents, generation of toxic ethanol metabolites and consequent activation of oxidative stress and inflammatory cascades [15,16,46,47,48,49]. In line with this view, the present observations indicate a high rate of abnormalities in GGT, a marker of oxidative stress and CRP, a biomarker of inflammation, especially in binge-type drinkers. GGT activation has previously been shown to coincide with the generation of superoxide ion, oxidation of lipoproteins and activation of a pro-inflammatory status in the body [48,49,50,51,52]. Elevation of blood neutrophils in alcoholic patients with recent drinking have also been shown to correlate with increased activities of serum liver enzymes [53]. Combinations of chronic and binge type drinking also aggravate hepatic inflammation and liver damage through upregulation of pro-inflammatory cytokines and endothelial-leukocyte adhesion molecules [54,55]. High levels of ethanol in circulation may also induce immunological responses to ethanol metabolite-specific neoantigens [56,57,58].

While here the overall biomarker responses appeared fairly similar between genders, it should be emphasized that some of the reported patterns were not consistent. The responses in ALT, lipid status and inflammation appeared more pronounced in men. It should, however, be noted that women show biomarker responses following smaller actual amounts of consumption. Based on previous literature women are also considered more vulnerable to alcohol-induced damage in the liver and central nervous system [7,14,59,60]. Stimulation of oxidative stress and inflammatory consequences of heavy drinking may also occur in a sex-dependent manner [16,51,61,62,63]. Therefore, further studies appear clearly warranted to compare the metabolic consequences of alcohol drinking in men and women with different alcohol preference types.

The present observations also support the view that a wide variety of lifestyle-related risk factors and their combinations may play a significant role in the metabolic consequences of alcohol intake in individuals with different alcohol drinking patterns and preference types. Previous studies have suggested that wine drinkers typically drink together with meals and also consume more vegetables [27,29]. The present data among male participants indicate a higher number of physical exercises in wine drinkers than in those preferring other types of alcoholic beverages. Binge-type drinking was also found here to be a frequently co-occurring behavior with smoking [64,65]. Thus, the synergistic effects between alcohol use and other lifestyle risk factors need further consideration when evaluating the metabolic and inflammatory consequences of alcohol drinking [66,67,68,69,70]. Future studies should also address the possibility whether for example those preferring beer or spirits together with smoking would be at a higher risk for developing alcohol dependence and related medical disorders [1,2,3,4,5,71]. On the other hand. it may be assumed that interventions aimed at reducing the total number of high-risk factors of lifestyle could be useful in harm reduction related to alcohol drinking.

The strengths of this study are the large number of participants and separate assessments for both sexes. The questionnaire used for collecting data covered the evaluation of a wide variety of aspects related to alcohol drinking, including the quantities of regular alcohol consumption, the types of alcoholic beverages preferred and the frequencies of binge drinking occasions, which should allow a comprehensive assessment of the associations between drinking patterns and health outcomes. Although at an individual level variation in drinking patterns may occur over time, the data collected here from the past one year showed a strong correlation with the corresponding data from the past one week prior to sampling (r_s_ = 0.76, *p* < 0.0005) indicating that the findings represent relatively consistent traits in alcohol consummatory behavior. The stability of the biomarkers should also be sufficient to reflect the current status of liver dysfunction, lipid status and inflammation at the time of blood sampling. Various other factors related to lifestyle, such as smoking, indices of overweight, physical activity or coffee consumption were also considered in the analyses. Nevertheless, our study has potential limitations. Self-report data may lead to overestimation of the proportion of abstainers and underestimation of the true dose–response associations [72]. Although in this work we reached relatively high participation rates, it is possible that the generalizability of the findings could suffer from non-response-patterns. Previous studies have indicated that especially high-risk drinkers are overrepresented in non-responders of population-based studies [73]. However, we feel that this would more likely dilute our findings than cause overestimates in the observed associations. At this time, we also cannot rule out the possibility of differences in self-report data between binge drinkers and regular alcohol consumers since self-reports represent a memory-dependent information channel. The cross-sectional setting of the survey hampering the assessment of causal inferences can also be kept as a shortcoming of this study.

Nevertheless, our study demonstrates that the type of alcohol preferred in consumption plays a significant role in modulating the effects of alcohol drinking on health. There may also be distinct differences in alcohol consumers with or without heavy episodic drinking. These findings should be implicated for formulation of low-risk drinking guidelines and in public health policies [6,22]. In clinical practice, the binge-type patterns of drinking deserve further attention in the development of specific questionnaires for identifying alcohol use disorders [15,35]. Measurements of key biomarkers can be recommended as complementary tools to obtain information on the individual status of inflammation and oxidative stress and for prediction of lifestyle-associated adverse health outcomes [6,14,48,74,75,76,77].

## Figures and Tables

**Table 1 nutrients-14-04529-t001:** Main characteristics of the study population, as classified according to the characteristics of alcohol consumption.

Men, *N* = 10 626	Abstainers	Alcohol Consumers	Type of Alcohol Preferred				
			Wine	Beer	Cider or long drink	Spirits	Mixed
*n* (% of *N*)	3046 (28.7)	7580 (71.3)	981 (9.2)	3691 (34.7)	144 (1.4)	1426 (13.4)	1338 (12.6)
Age, years, mean ± SD	51.9 ± 14.1	49.0 ± 13.2 ^a^	53.2 ± 12.6 ^b,d^	45.5 ± 12.9 ^a,d^	46.2 ± 14.1 ^a,d^	54.0 ± 11.9 ^a,d^	50.5 ± 12.9 ^b^
Alcohol consumption, g/day, mean ± SD	0.0 ± 0.0	18.1 ± 19.4	14.0 ± 13.8 ^d^	18.8 ± 20.0 ^d^	11.0 ± 12.7 ^d^	18.1 ± 19.9 ^e^	19.9 ± 20.9 ^d^
Average level of drinking, n (%)							
moderate	N/A	5815 (76.7)	835 (85.1) ^d^	2755 (74.6) ^f^	128 (88.9) ^d^	1088 (76.3)	1009 (75.4)
heavy	N/A	1765 (23.3)	146 (14.9)	936 (25.4)	16 (11.1)	338 (23.7)	329 (24.6)
Drinking pattern, n (%)							
regular	N/A	5025 (66.3)	755 (77.0) ^d^	2334 (63.2) ^e^	116 (80.6) ^d^	896 (62.8) ^f^	924 (69.1) ^f^
binge	N/A	2555 (33.7)	226 (23.0)	1357 (36.8)	28 (19.4)	530 (37.2)	414 (30.9)
BMI, mean ± SD	27.4 ± 4.3	27.1 ± 4.0 ^b^	27.2 ± 3.9	26.7 ± 3.9 ^a,d^	26.9 ± 3.9	28.2 ± 4.5 ^a,d^	27.1 ± 3.6 ^c^
Waist circumference, cm, mean ± SD	96.4 ± 12.3	95.8 ± 11.5 ^c^	96.3 ± 11.2	94.4 ± 11.4 ^a,d^	94.5 ± 11.2	99.0 ± 12.3 ^a,d^	95.7 ± 10.5
Smoking, cigarettes/day, mean ± SD	4.1 ± 8.6	5.2 ± 9.1 ^a^	2.5 ± 6.3 ^a,d^	6.2 ± 9.7 ^a,d^	4.6 ± 10.0	5.5 ± 9.5 ^a,d^	3.9 ± 8.3 ^f^
Coffee, cups/day, mean ± SD	4.7 ± 3.5	4.6 ± 3.1 ^c^	4.0 ± 2.8 ^a,d^	4.8 ± 3.3 ^d^	4.5 ± 3.4	4.8 ± 3.2 ^d^	4.2 ± 2.8 ^a,e^
Physical activity, number of exercises per week, mean ± SD	2.5 ± 2.4	2.3 ± 2.1 ^b^	2.6 ± 2.0 ^e^	2.2 ± 2.0 ^a^	1.6 ± 1.8 ^c,f^	2.4 ± 2.4	2.4 ± 1.9
Women, *N* = 11 806							
*n* (% of *N*)	5174 (43.8)	6632 (56.2)	2278 (19.3)	2197 (18.6)	429 (3.6)	600 (5.1)	1128 (9.6)
Age, years, mean ± SD	49.6 ± 14.1	46.9 ± 12.7 ^a^	49.9 ± 12.9 ^d^	43.4 ± 11.5 ^a,d^	43.8 ± 12.9 ^a,d^	49.9 ± 12.9 ^d^	47.5 ± 12.5 ^a^
Alcohol consumption, g/day, mean ± SD	0.0 ± 0.0	8.6 ± 9.1	8.1 ± 9.4	8.7 ± 8.9	5.7 ± 5.3 ^d^	8.5 ± 10.4	10.5 ± 8.7 ^d^
Average level of drinking, n (%)							
moderate	N/A	5284 (79.7)	1887 (82.8) ^e^	1715 (78.1)	386 (90.0) ^d^	490 (81.7)	806 (71.5) ^d^
heavy	N/A	1348 (20.3)	391 (17.2)	482 (21.9)	43 (10.0)	110 (18.3)	322 (28.5)
Drinking pattern, n (%)							
regular	N/A	5873 (88.6)	2073 (91.0) ^e^	1872 (85.2) ^d^	409 (95.3) ^d^	515 (85.8) ^f^	1004 (89.0)
binge	N/A	759 (11.4)	205 (9.0)	325 (14.8)	20 (4.7)	85 (14.2)	124 (11.0)
BMI, mean ± SD	27.2 ± 5.5	26.0 ± 4.8 ^a^	26.1 ± 4.8 ^a^	25.6 ± 4.7 ^a,d^	26.4 ± 5.3 ^b^	27.0 ± 5.1 ^d^	26.1 ± 4.9 ^a^
Waist circumference, cm, mean ± SD	85.6 ± 13.7	83.2 ± 12.5 ^a^	83.5 ± 12.3 ^a^	82.3 ± 12.1 ^a,d^	82.9 ± 13.7 ^a^	85.8 ± 13.6 ^d^	83.2 ± 12.5 ^a^
Smoking, cigarettes/day, mean ± SD	1.8 ± 5.0	2.7 ± 6.0 ^a^	1.6 ± 4.5 ^d^	4.2 ± 7.1 ^a,d^	2.9 ± 6.5 ^a^	3.4 ± 7.3 ^a,e^	1.9 ± 4.8 ^d^
Coffee, cups/day, mean ± SD	3.7 ± 2.6	3.7 ± 2.4	3.3 ± 2.2 ^a,d^	4.1 ± 2.6 ^a,d^	3.8 ± 2.6	3.8 ± 2.6	3.6 ± 2.2 ^f^
Physical activity, number of exercises per week, mean ± SD	2.5 ± 2.1	2.5 ± 2.0	2.6 ± 2.1	2.4 ± 2.0	2.3 ± 1.8	2.3 ± 1.9	2.6 ± 1.8 ^f^

BMI, body mass index; N/A, not applicable; ^a^
*p* < 0.001 ^b^
*p* < 0.01 ^c^
*p* < 0.05 for comparisons between abstainers vs. alcohol drinkers, ^d^
*p* < 0.001 ^e^
*p* < 0.01 ^f^
*p* < 0.05 for alcohol consumers in general vs. subgroups preferring different types of alcoholic beverages.

**Table 2 nutrients-14-04529-t002:** The percentages of values exceeding the upper normal limits of each laboratory test in the study groups.

Men	Abstainers	Alcohol Consumers	Type of Alcohol Preferred				
			Wine	Beer	Cider Or Long Drink	Spirits	Mixed
	%	%	%	%	%	%	%
GGT ≥ 60U/L	11.8	18.9 ^a^	17.3 ^a^	18.4 ^a^	11.8 ^f^	23.1 ^a,d^	17.9 ^a^
ALT ≥ 50 U/L	9.3	13.1 ^b^	11.0	12.6 ^c^	16.1	14.7 ^b^	13.8 ^b^
Cholesterol ≥ 5 mmol/L	63.0	69.3 ^a^	65.7 ^f^	67.9 ^a^	61.1 ^f^	73.4 ^a,e^	72.6 ^a,f^
HDL ≤ 1 mmol/L	25.4	15.2 ^a^	14.2 ^a^	14.5 ^a^	20.1	17.4 ^a,f^	14.9 ^a^
LDL ≥ 3 mmol/L	61.4	65.3 ^b^	60.8 ^f^	63.8	65.6	68.8 ^a,f^	69.1 ^a,f^
Triglycerides ≥ 1.7 mmol/L	34.2	35.1	31.9	33.8	36.8	39.6 ^a,e^	35.7
CRP-hs ≥ 3 mg/L	19.0	18.3	15.5 ^c,f^	17.1 ^c^	17.9	23.8 ^a,d^	17.9
MCV > 96 fL	11.4	18.2 ^b^	16.7	19.4 ^b^	10.0	20.4 ^b^	15.6
Women							
GGT ≥ 40U/L	10.5	12.3 ^b^	11.8	12.1 ^c^	7.9 ^e^	16.7 ^a,e^	12.9 ^c^
ALT ≥ 35 U/L	8.2	9.4	9.8	8.9	7.5	12.2 ^c^	8.2
Cholesterol ≥ 5 mmol/L	67.1	62.1 ^a^	63.4 ^b^	58.7 ^a,e^	58.7 ^a^	66.3 ^f^	64.9
HDL ≤ 1.2 mmol/L	20.0	11.7 ^a^	10.7 ^a^	12.0 ^a^	17.0 ^d^	14.8 ^b,f^	9.4 ^a,f^
LDL ≥ 3 mmol/L	59.4	52.9 ^a^	53.8 ^a^	50.4 ^a^	52.2	56.5	54.7 ^c^
Triglycerides ≥ 1.7 mmol/L	21.0	15.1 ^a^	14.6 ^a^	15.0 ^a^	16.3 ^c^	19.3 ^e^	13.7 ^a^
CRP-hs ≥ 3 mg/L	24.0	20.4 ^a^	20.2 ^a^	18.7 ^a^	22.4	26.5 ^d^	20.2 ^b^
MCV > 96 fL	17.1	18.8	21.5	15.5	0.0	28.6	20.0

^a^*p* < 0.001 ^b^
*p* < 0.01 ^c^
*p* < 0.05 for comparisons with abstainers, ^d^
*p* < 0.001 ^e^
*p* < 0.01 ^f^
*p* < 0.05 for comparisons with the total group of alcohol consumers. For MCV, n = 1715 (1115 men, 600 women). ALT, alanine aminotransferase; CRP, C-reactive protein; GGT, gamma glutamyl transferase; HDL, high density lipoprotein; LDL, low density lipoprotein; MCV, mean corpuscular volume.

**Table 3 nutrients-14-04529-t003:** Odds ratios for abnormal biomarker status in alcohol consumers according to dominant alcoholic beverage type, as adjusted for age, BMI, physical activity, smoking and coffee consumption.

		Men		Women	
		Odds Ratio (95% CI)	*p*	Odds Ratio (95% CI)	*p*
GGT	Abstainers	1.0		1.0	
	Wine	1.77 (1.44 to 2.18)	<0.001	1.34 (1.14 to 1.58)	<0.001
	Beer	1.94 (1.67 to 2.25)	<0.001	1.68 (1.41 to 1.99)	<0.001
	Cider or long drink	1.16 (0.69 to 1.95)	0.584	0.95 (0.66 to 1.37)	0.771
	Spirits	2.06 (1.73 to 2.46)	<0.001	1.77 (1.39 to 2.26)	<0.001
	Mixed	1.87 (1.55 to 2.25)	<0.001	1.64 (1.34 to 2.01)	<0.001
ALT	Abstainers	1.0		1.0	
	Wine	1.48 (0.96 to 2.26)	0.076	1.35 (1.02 to 1.80)	0.039
	Beer	1.37 (1.03 to 1.81)	0.028	1.37 (1.03 to 1.83)	0.032
	Cider or long drink	1.81 (0.62 to 5.25)	0.278	0.92 (0.36 to 2.34)	0.852
	Spirits	1.77 (1.27 to 2.48)	<0.001	1.57 (1.03 to 2.39)	0.035
	Mixed	1.68 (1.18 to 2.40)	0.004	1.13 (0.77 to 1.65)	0.526
Cholesterol	Abstainers	1.0		1.0	
	Wine	1.14 (0.98 to 1.33)	0.092	0.86 (0.77 to 0.96)	0.008
	Beer	1.44 (1.29 to 1.60)	<0.001	0.93 (0.83 to 1.04)	0.194
	Cider or long drink	1.01 (0.71 to 1.42)	0.967	0.96 (0.78 to 1.18)	0.685
	Spirits	1.46 (1.27 to 1.68)	<0.001	0.93 (0.77 to 1.13)	0.471
	Mixed	1.68 (1.46 to 1.94)	<0.001	1.04 (0.90 to 1.20)	0.607
HDL-cholesterol	Abstainers	1.0		1.0	
	Wine	0.50 (0.41 to 0.61)	<0.001	0.55 (0.47 to 0.64)	<0.001
	Beer	0.51 (0.45 to 0.58)	<0.001	0.59 (0.51 to 0.69)	<0.001
	Cider or long drink	0.77 (0.50 to 1.17)	0.219	0.90 (0.69 to 1.17)	0.410
	Spirits	0.55 (0.47 to 0.65)	<0.001	0.65 (0.51 to 0.84)	<0.001
	Mixed	0.52 (0.43 to 0.62)	<0.001	0.46 (0.37 to 0.57)	<0.001
LDL-cholesterol	Abstainers	1.0		1.0	
	Wine	0.99 (0.82 to 1.19)	0.908	0.81 (0.72 to 0.92)	0.001
	Beer	1.16 (1.02 to 1.32)	0.025	0.89 (0.78 to 1.01)	0.067
	Cider or long drink	1.22 (0.72 to 2.09)	0.456	0.77 (0.54 to 1.10)	0.154
	Spirits	1.30 (1.09 to 1.54)	0.003	0.82 (0.66 to 1.02)	0.082
	Mixed	1.45 (1.22 to 1.73)	<0.001	0.92 (0.78 to 1.09)	0.337
Triglycerides	Abstainers	1.0		1.0	
	Wine	0.98 (0.83 to 1.15)	0.777	0.72 (0.62 to 0.83)	<0.001
	Beer	1.12 (1.00 to 1.25)	0.048	0.93 (0.80 to 1.08)	0.339
	Cider or long drink	1.23 (0.85 to 1.77)	0.272	0.95 (0.72 to 1.25)	0.714
	Spirits	1.10 (0.95 to 1.26)	0.203	0.90 (0.72 to 1.13)	0.347
	Mixed	1.19 (1.03 to 1.37)	0.020	0.72 (0.59 to 0.87)	<0.001
CRP	Abstainers	1.0		1.0	
	Wine	0.82 (0.67 to 1.00)	0.052	0.97 (0.85 to 1.11)	0.629
	Beer	1.02 (0.89 to 1.17)	0.777	0.95 (0.83 to 1.10)	0.512
	Cider or long drink	1.12 (0.71 to 1.76)	0.634	1.04 (0.80 to 1.35)	0.778
	Spirits	1.11 (0.94 to 1.30)	0.216	1.19 (0.96 to 1.47)	0.108
	Mixed	1.03 (0.86 to 1.23)	0.754	0.98 (0.82 to 1.16)	0.809
MCV	Abstainers	1.0		1.0	
	Wine	1.60 (0.89 to 2.89)	0.120	1.30 (0.69 to 2.48)	0.417
	Beer	1.80 (1.15 to 2.84)	0.011	0.87 (0.48 to 1.58)	0.648
	Cider or long drink	0.77 (0.09 to 6.49)	0.813	NA	
	Spirits	1.89 (1.09 to 3.27)	0.024	1.85 (0.79 to 4.33)	0.155
	Mixed	1.27 (0.72 to 2.24)	0.403	1.27 (0.57 to 2.80)	0.557

For abbreviations and marker cut-offs, see Table 2. NA, not applicable.

**Table 4 nutrients-14-04529-t004:** Percentages of abnormal values in different biomarkers in subgroups with different patterns of alcohol consumption. *P* values refer to trends in distribution of observations.

		Average Level of Drinking				
		Moderate		Heavy		
		Drinking Pattern		Drinking Pattern		
Men		Regular	Binge	Regular	Binge	*p*
		%	%	%	%	
GGT ≥ 60 U/L	wine	13.5	21.3	27.9	34.1	<0.001
	beer	12.2	16.6	33.3	33.3	<0.001
	spirits	16.4	25.7	31.3	38.8	<0.001
	mixed	11.5	22.0	29.6	32.2	<0.001
ALT ≥ 50 U/L	wine	8.7	14.6	20.0	14.7	0.082
	beer	10.0	11.7	14.3	20.2	<0.001
	spirits	11.1	13.2	19.4	27.7	<0.001
	mixed	10.9	16.9	20.0	17.9	0.037
Cholesterol ≥ 5 mmol/L	wine	64.3	62.4	77.0	75.3	0.017
	beer	66.0	67.1	77.2	70.1	0.004
	spirits	71.1	73.3	83.3	76.9	0.019
	mixed	73.5	72.0	72.2	70.1	0.311
HDL ≤ 1 mmol/L	wine	15.9	12.1	16.4	2.4	0.003
	beer	17.8	12.1	10.4	8.6	<0.001
	spirits	19.0	17.0	21.9	10.7	0.012
	mixed	17.2	9.5	13.9	11.7	0.024
LDL ≥ 3 mmol/L	wine	59.3	63.8	64.3	65.5	0.244
	beer	64.4	60.7	68.7	63.5	0.969
	spirits	68.6	68.0	68.5	70.4	0.710
	mixed	70.6	67.7	74.6	61.1	0.097
Triglycerides ≥ 1.7 mmol/L	wine	31.0	35.5	39.3	28.2	0.732
	beer	32.1	32.6	43.3	36.3	0.003
	spirits	37.1	39.9	45.8	44.6	0.017
	mixed	34.4	32.5	33.9	44.9	0.016
CRP ≥ 3mg/L	wine	14.7	20.6	20.0	9.6	0.854
	beer	15.2	15.6	20.3	23.3	<0.001
	spirits	20.0	24.0	31.3	33.2	<0.001
	mixed	17.2	19.4	18.9	18.7	0.521
Women						
GGT ≥ 40 U/L	wine	10.8	14.5	16.0	15.6	0.005
	beer	8.9	11.0	21.2	25.9	<0.001
	spirits	14.7	19.0	19.4	30.2	0.011
	mixed	11.2	13.5	16.2	18.4	0.010
ALT ≥ 35 U/L	wine	9.3	12.0	14.7	6.4	0.511
	beer	8.1	4.1	11.7	16.0	0.014
	spirits	11.1	21.7	10.0	15.0	0.612
	mixed	8.4	5.0	8.9	6.5	0.807
Cholesterol ≥ 5 mmol/L	wine	63.0	53.0	72.1	57.4	0.435
	beer	59.0	48.5	70.0	46.0	0.396
	spirits	67.2	64.3	62.7	65.1	0.506
	mixed	67.0	56.8	63.0	55.2	0.026
HDL ≤ 1.2 mmol/L	wine	11.6	7.2	7.4	6.6	0.007
	beer	12.9	11.0	8.5	10.1	0.036
	spirits	14.7	19.0	14.9	11.6	0.787
	mixed	10.1	10.8	9.8	1.1	0.062
LDL ≥ 3 mmol/L	wine	54.0	35.4	61.1	48.4	0.936
	beer	52.4	40.2	53.2	37.1	0.005
	spirits	61.1	44.1	39.6	52.8	0.015
	mixed	57.6	29.0	53.0	48.6	0.079
Triglycerides ≥ 1.7 mmol/L	wine	14.6	14.5	14.5	14.8	0.982
	beer	13.9	12.5	19.1	19.6	0.006
	spirits	17.9	11.9	22.4	37.2	0.009
	mixed	13.5	5.4	15.3	13.8	0.670
CRP ≥ 3mg/L	wine	20.8	19.5	18.6	14.9	0.097
	beer	17.4	14.2	25.2	21.8	0.007
	spirits	24.9	37.5	23.9	37.2	0.154
	mixed	19.6	10.8	23.7	20.7	0.321

For abbreviations, see Table 2.

## Data Availability

THL Biobank administrates and grants access to the FINRISK data to research projects that are of high scientific quality and impact, are ethically conducted, and that correspond with the research areas of THL Biobank. All data are available for application at https://thl.fi/en/web/thl-biobank/for-researchers/sample-collections/the-national-finrisk-study-1992-2012 (accessed on 12 May 2022). The name of dataset is the National FINRISK Study 1992-2012. Interested researchers can replicate our study findings in their entirety by directly obtaining the data and following the protocol in the Methods section. The authors did not have any special access privileges that others would not have. More information: finriski(at)thl.fi.

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
