# Peer review of "Alcohol Drinking Patterns and Laboratory Indices of Health: Does Type of Alcohol Preferred Make a Difference?"

_nutrients, 2022, doi:10.3390/nu14214529_

Round 1
Reviewer 1 Report
The submission provides an analysis of a large cross-sectional national survey. Participants were assessed on various behaviors, including alcohol consumption and physical activity. Anthromorphic measures were obtained in a clinical evaluation. Blood chemistry was also assessed. Drinking was categorized based on patterns of consumption and dominant alcoholic beverage type. Unadjusted associations between various categories of drinking and biomarker status constituted the main approach to analyses. Analyses were sex-specific. The results point to several biomarkers that differ among types of drinkers.
This work can be a valuable contribution to the literature. Evaluation of metabolic consequences or differences among drinking patterns and types of alcohol consumed is often a challenge in many studies, and the question of its importance to consider is a shadow on the conclusions of many epi studies of alcohol. This work can help shed light on that.
There are a number of considerations I offer to the authors and editor to consider.
1. Causal inference. In the discussion, phrases such as the effects and may lead to significant variation hold a strong causal connotation. The evidence for causation is now well laid out and the actual causal effect sizes are not established by the current analyses. The analyses appear to provide raw unadjusted values. This can be useful for the literature. At the same time it highlights the analyses aren't well structured for causal estimation.
The authors do present a very good section on many of the potential confounders, which should justify the very deliberate and justified use of causal inferences. I would also suggest considering additional confounders among the alcohol preference types.
This was the most substantial consideration. The remaining comments are minor.
2. Low-moderate Drinking Benefits. The introduction addresses some of the older literature based on the prior generation of epi data on the topic. Here the authors might clarify what levels of drinking were considered optimal based on that literature and incorporate a recognition that newer literature (including papers by Stockwell et al. which are cited elsewhere) has cast some doubt on some of the earlier work.
3. Low-level drinking. The paper includes three classes (moderate and heavy). Where there no level/rare drinkers? If so, does the inclusion of these with regular drinkers wash out or strengthen the patterns observed? Similarly, how were daily binge drinkers handled? How did their pooling with more normalized drinking patterns among the heavy drinkers impact the reported patterns?
4. Measurement. More information on how the alcohol patterns were assessed should be provided. Twelve months were assessed. Was that the reference period for all the items? Was the period broken up such as in a timeline follow-back approach?
A discussion of match between how stable each of the biomarkers are and how much variation there might be in the drinking patterns would be appropriate. For example, if someone heavily binged in the prior month but not in the 11 months before that, in what ways would that impact the variance and patterns we see in the results?
5. Random sampling was used for ascertainment. What was the participation rate? Is there evidence of potential biases because of non-response/participation patterns that might relate to the findings or limits on generalizability?
6. Significance. Report all P-values even if they are not below a set alpha. Setting alpha is also a practice that has received much attention and is now commonly not recommended by statisticians who publish on this topic.
7. Test for trend. Where tests for trends are used clearly justify their use in the methods. In table 3 it is unclear what trend is being evaluated and if that is appropriate, as there are no latent gradients/scales over which a trend can be evaluated. (A model test such as likelihood ratio test might work, or a chi-square test).
8. Exploratory Patterns. The research is exploratory, and the modeling can be acceptable when presented with discussion and conclusions that recognize that patterns should be replicated prior to being accepted. Otherwise, suggestions that multiple comparison corrections should be used would take on some weight.
9. Consistency between Men and Women. Some of the reported patterns were not consistent between the sexes, such as for CRP in Table 3. Avoid leaving the reader thinking the patterns might be consistent.
10. MCV Subsample. Please describe how the MCV subsample was obtained and the size and direction of likely biases that result from the approach that was taken.
11. Magnitudes. The results text does not address the magnitudes of the differences. The significance of differences is declared, and there is little presentation and discussion of the differences. Some of the proportions differ by large amounts and others less so. It would make the paper easier for the reader if the authors noted which differences were large and which were smaller.
Author Response
The submission provides an analysis of a large cross-sectional national survey. Participants were assessed on various behaviors, including alcohol consumption and physical activity. Anthromorphic measures were obtained in a clinical evaluation. Blood chemistry was also assessed. Drinking was categorized based on patterns of consumption and dominant alcoholic beverage type. Unadjusted associations between various categories of drinking and biomarker status constituted the main approach to analyses. Analyses were sex-specific. The results point to several biomarkers that differ among types of drinkers.
This work can be a valuable contribution to the literature. Evaluation of metabolic consequences or differences among drinking patterns and types of alcohol consumed is often a challenge in many studies, and the question of its importance to consider is a shadow on the conclusions of many epi studies of alcohol. This work can help shed light on that.
There are a number of considerations I offer to the authors and editor to consider.
- Causal inference.In the discussion, phrases such as the effects and may lead to significant variation hold a strong causal connotation. The evidence for causation is now well laid out and the actual causal effect sizes are not established by the current analyses. The analyses appear to provide raw unadjusted values. This can be useful for the literature. At the same time it highlights the analyses aren't well structured for causal estimation.
The authors do present a very good section on many of the potential confounders, which should justify the very deliberate and justified use of causal inferences. I would also suggest considering additional confounders among the alcohol preference types.
This was the most substantial consideration. The remaining comments are minor.
We wish to thank the reviewer for the positive overall assessment and highly useful suggestions concerning our work. We agree with the view that due to the observational and cross-sectional nature of the work and lack of follow-up data it is difficult to demonstrate causal relationships. In the revised manuscript, we have now sharpened the presentation by de-emphasizing the causal estimations, as appropriate. We have also conducted additional analyses of the differences by using a number of potential confounders, as recommended. An additional table showing the data from comparisons between alcohol consumers and abstainers, as adjusted for age, BMI, physical activity, smoking and coffee consumption, has been added (Table 3 of the revised manuscript). In the Discussion Section, we have also further emphasized the role of possible interactions between alcohol drinking and other lifestyle-related factors in mediating inflammation and the adverse metabolic consequences.
- Low-moderate Drinking Benefits.The introduction addresses some of the older literature based on the prior generation of epi data on the topic. Here the authors might clarify what levels of drinking were considered optimal based on that literature and incorporate a recognition that newer literature (including papers by Stockwell et al. which are cited elsewhere) has cast some doubt on some of the earlier work.
Based on the reviewer’s recommendation we have now sharpened the Introduction and the Discussion Sections by further emphasizing the existing controversies in defining safe or optimal levels of drinking. The Introduction has also been strengthened by including an additional reference presenting a systematic meta-regression analysis of this topic (Stockwell et al J Stud Alcohol Drugs 77, 185-, 2016).
- Low-level drinking. The paper includes three classes (moderate and heavy). Where there no level/rare drinkers? If so, does the inclusion of these with regular drinkers wash out or strengthen the patterns observed? Similarly, how were daily binge drinkers handled? How did their pooling with more normalized drinking patterns among the heavy drinkers impact the reported patterns?
In the present work, we have used the data on total regular alcohol consumption to categorize the material according to current NIAAA recommendations as follows: 1. abstainers, 2. moderate drinkers (≤ 7 drinks per week for women or ≤ 14 drinks per week for men) or 3. heavy drinkers, who exceeded 7 drinks per week (women) or 14 drinks per week (men). We also divided the population according to the international recommendations for classifying binge drinking. We feel that this approach from a nationally representative population-based sample with a wide range of alcohol consumption should provide a solid basis for obtaining a comprehensive view on the differences between the various levels and patterns of drinking and efficient comparisons of the data with earlier literature in this field. In this material subjects with very high intensities of drinking, such as daily binge drinking, were rare. In preliminary analyses, which were restricted to only those with moderate to high alcohol drinking levels we observe even stronger associations. However, we feel that such subgroup specific analyses should be a subject of further studies and consist a separate communication. In the additional analyses carried out on the correlations between the actual number of binge drinking episodes, the strongest associations were noted between GGT (rs = 0.234, p < 0.001) in men preferring spirit and between MCV in men (rs = 0.258, p < 0.001) and women (rs = 0.270, p < 0.001) preferring beer. This information has now also been given in the revised manuscript.
- Measurement.More information on how the alcohol patterns were assessed should be provided. Twelve months were assessed. Was that the reference period for all the items? Was the period broken up such as in a timeline follow-back approach?
A discussion of match between how stable each of the biomarkers are and how much variation there might be in the drinking patterns would be appropriate. For example, if someone heavily binged in the prior month but not in the 11 months before that, in what ways would that impact the variance and patterns we see in the results?
We have now provided a more detailed account on the assessment of alcohol drinking patterns. In the present material we have used data from the questionnaire covering the past 12 months of drinking as the time frame of assessing drinking habits and the frequency of binge drinking. The quantitative data on alcohol drinking habits was also recorded in detail from the past one week and showed a good correlation with the corresponding data obtained from the past one year (Spearman’s rho =0.76, p < 0.0005) indicating relatively consistent traits in alcohol-drinking behaviour. The laboratory tests used here were analyzed with high-precision methods which minimize the possibility of errors in analytical procedures. The stability of the biomarkers with normalization times ranging from several days to several weeks (liver enzymes) or even months (MCV) should also be sufficient to justify the conclusions reached on the status of liver dysfunction, lipid status and inflammation at the time of blood sampling and data collection. The above views have now been covered in the Discussion Section of the revised manuscript and the possibility of variations in drinking patterns over time has also been mentioned among the limitations of the study.
- Random sampling was used for ascertainment. What was the participation rate? Is there evidence of potential biases because of non-response/participation patterns that might relate to the findings or limits on generalizability?
We have now provided additional information on the study participation rates, which varied between the different study years (1997-2007) from 52.6% to 73.4% (as percentages of the individuals who both filled the questionnaire and attended the medical examinations). In international comparisons of population-based studies, the participation rates of the FINRISK Study have been shown to be high (Borodulin et al Int J Epidemiol 2018, 47, 696-). Based on previous experience from population studies non-participants seem to be overrepresented in high-risk drinkers (Tolonen et al. Eur J Epidemiology 2010;25:69-76; Torvik et al 2012 Psych Epidemiol 2012 47, 805-). However, we believe that such non-responsiveness in the present material would more likely dilute our findings than cause overestimates in the observed associations. In the revised manuscript, we have now covered the above views in the Discussion Section.
- Significance. Report all P-values even if they are not below a set alpha. Setting alpha is also a practice that has received much attention and is now commonly not recommended by statisticians who publish on this topic.
Based on the reviewer’s recommendation we have now listed all P-values in Tables 3-4 of the revised manuscript. In tables 1 and 2, which originally were very busy, only the P-values denoting the different levels of significance are given to keep the tables reader-friendly.
- Test for trend. Where tests for trends are used clearly justify their use in the methods. In table 3 it is unclear what trend is being evaluated and if that is appropriate, as there are no latent gradients/scales over which a trend can be evaluated. (A model test such as likelihood ratio test might work, or a chi-square test).
In the revised manuscript, we have now clarified the presentation on the use of the tests for trends (Statistical Methods and Results Section). Table 3 (=table 4 of the revised manuscript) shows the distribution of findings exceeding the cut-offs for the biomarkers in ordered alcohol-consuming study groups and thus a chi-square test for trend should be an appropriate method here.
- Exploratory Patterns. The research is exploratory, and the modeling can be acceptable when presented with discussion and conclusions that recognize that patterns should be replicated prior to being accepted. Otherwise, suggestions that multiple comparison corrections should be used would take on some weight.
In the present work, statistical comparisons were performed exploratively without correction for multiple comparisons to reveal tendencies. We have now further clarified this approach in the revised manuscript and emphasized also the views brought up by the reviewer in the discussion and conclusions on the observed patterns. The need for future studies in this field has also been acknowledged.
- Consistency between Men and Women.Some of the reported patterns were not consistent between the sexes, such as for CRP in Table 3. Avoid leaving the reader thinking the patterns might be consistent.
The reviewer raises an important point. We have now modified the presentation to further emphasize the differences observed between sexes.
- MCV Subsample.Please describe how the MCV subsample was obtained and the size and direction of likely biases that result from the approach that was taken.
In the present study, blood cell counts were determined from only one out of six regional subsamples of the FINRISK Study. In the other study locations MCV analyses (which would have required prompt locally performed assays from fresh whole blood samples) could not be carried out due to both practical and financial reasons, including lack of immediate access to appropriate hematology analyzers. All other laboratory parameters were based on assays from serum specimens, which were processed at the collection site and subsequently shipped to a centralized laboratory for high-throughput analyses by experienced laboratory personnel. In the revised manuscript we have now given additional information on MCV analyses and also mentioned the possible limitations of the smaller number of observations in this subsample in the Discussion Section of the revised manuscript.
- Magnitudes. The results text does not address the magnitudes of the differences. The significance of differences is declared, and there is little presentation and discussion of the differences. Some of the proportions differ by large amounts and others less so. It would make the paper easier for the reader if the authors noted which differences were large and which were smaller.
Based on the reviewer’s recommendation we have now placed more emphasis more also on the magnitudes of the differences observed. In line with this suggestion, the P values of all observations even if they are not below a set alpha have also been included in Tables 3-4 of the revised manuscript (see also comment no 6).
Reviewer 2 Report
This was an innovative study with a large database. All sections were adequately described but I do have an issue with your methods:
(1) Why did you not control for the non-alcohol consumption factors in your analyses? The physical exercise, coffee consumption, smoking cigarettes, BMI, etc. While you discuss them it would have been much more productive to see if binge-drinking beer preference male drinkers differed from the other groups (abstainers, wine drinkers, etc.) after controlling for the above factors. Maybe. Maybe not. But you did not do that. This is a major factor in your study.
(2) Did you include medications that these people were taking in your analyses? I did not see that.
Author Response
This was an innovative study with a large database. All sections were adequately described but I do have an issue with your methods:
- Why did you not control for the non-alcohol consumption factors in your analyses? The physical exercise, coffee consumption, smoking cigarettes, BMI, etc. While you discuss them it would have been much more productive to see if binge-drinking beer preference male drinkers differed from the other groups (abstainers, wine drinkers, etc.) after controlling for the above factors. Maybe. Maybe not. But you did not do that. This is a major factor in your study.
We thank the reviewer for the positive overall assessment and useful suggestions concerning our work. In the revised manuscript we have now further strengthened the presentation by conducting additional analyses to compare alcohol consumers and abstainers as adjusted for age, BMI, physical activity, smoking and coffee consumption. A new table describing this data has also been added (Table 3 of the revised manuscript). In the Discussion Section of the revised manuscript we have also further emphasized the possible interactions between alcohol use and other unfavourable lifestyle factors in mediating inflammation and adverse metabolic consequences.
- Did you include medications that these people were taking in your analyses? I did not see that.
In the present health survey the participants were apparently healthy individuals and conditions such as history or current clinical signs of liver disease, ischemic heart or brain disease or active infection at the time of blood sampling were excluded. We have now also analyzed the data collected on the current use of medications prior to sampling and included this information in the Methods Section of the revised manuscript. In the total population, 22.8% had reported occasional use of analgesics. The other medications used were: antidepressants (4.7%), sedatives (5.9%), inhaled asthma medication (5.7%), antihistamines (3.0%) and low-dose acetylsalicylic acid (10.2%). However, in light of current knowledge on the possible interactions between the present study parameters and the medication used we do not expect any significant bias due to such considerations.